# THE ROLE OF LEARNING AND MEMORIZATION IN RELABELING-BASED UNLEARNING FOR LLMS

## ABSTRACT

This work studies how the nature of a response generated by a large language model (LLM) impacts the efficiency of *relabeling-based unlearning*, a common unlearning technique that trains the model to fit an "*unlearn*" set (i.e., a dataset that we wish the model to unlearn) with alternative responses to prevent it from generating unwanted outputs that align with the unlearn set. We distinguish between two different ways LLMs can generate undesirable outputs: **learning-based generation**, where the model learns an underlying *rule* connecting the input and the response (e.g., social stereotypes), and **memorization-based generation**, where the model memorizes specific information about a given input (e.g., private information like a phone number). We demonstrate that relabeling-based unlearning can be detrimental to the model performance when the undesirable outputs are generated based on learning-based generation whereas it is more effective with memorization-based generation. We provide theoretical justifications for this through the lens of hypothesis testing, showing that memorization-based hypotheses are more stable in the presence of *"fabricated evidence"* that contradicts the hypothesis' prediction and more flexible to produce alternative responses. Our empirical results further support our findings by showing a clear performance gap in relabeling-based unlearning under these two types of data generation mechanisms.

## 1 INTRODUCTION

Large language models (LLMs) have shown remarkable capability to generate complex, human-like text by being pretrained on massive amount of data. However, this training process may also result in undesirable model behaviors with serious safety risks, such as privacy leakage, social bias, and creation of harmful content (Carlini et al., 2021; Huang et al., 2022; Sandbrink, 2023). To address these risks, machine unlearning for LLMs has emerged as an increasingly growing field aimed at removing the influence of undesirable data from a model.

A primary objective of machine unlearning for LLMs (also referred to as LLM unlearning) is to prevent the model from producing undesirable outputs, represented by a data set known as the *unlearn* set (also known as the forget set), given relevant prompts without compromising the model's overall capability. Researchers have proposed various unlearning methods based on different frameworks (Jang et al., 2022; Zhang et al., 2024; Jiang et al., 2020; Li et al., 2024). Among these methods, a common approach is the *relabeling-based method* (Deeb & Roger, 2024; Maini et al., 2024; Eldan & Russinovich, 2023). The core idea for the relabeling-based method is to first create a new unlearn set with the same prompts as the original unlearn set but with alternative, harmless responses. The model is then trained with this new set being a part of the training data. By training the model to fit these new responses, we effectively encourage the model to predict the original prompts with alternative responses and then overwrite the undesirable responses.

While the field of LLM machine unlearning is receiving increasing attention, the understanding on what factors truly influence its difficulty and efficiency is relatively underexplored. This is a crucial area of study since the understanding of these factors allows us to gain deeper insights on how these unlearning methods work and help us develop more effective and reliable unlearning methods. Prior works in this area have identified several key factors that influence the unlearning efficiency including frequency of unlearn data in the training data (Krishnan et al., 2025), data

entanglement (Zhao et al., 2024a), robustness to parameter perturbation (Feng et al., 2025), how knowledge is encoded in the training data (Wu et al., 2025) and so on.

Following this line of work, our work studies what affects the unlearning difficulty and efficiency, specifically for the *relabeling-based unlearning method*. Our focus is a new factor: how the undesirable response is generated by the model after the initial training. In this work, we distinguish between two different ways of response generation: **learning-based generation**, where the model learns a general rule to connect the prompts to the response (e.g., social stereotypes that associate certain professionals with a particular gender), and **memorization-based generation**, where the model memorizes a specific response to given prompts (e.g., private information like a personal phone number).

In this work, we study how the nature of response generation (learning-based versus memorization-based) affects the unlearning efficiency for the relabeling-based method. The key contributions of this work are highlighted as follows:

- We propose a hypothesis testing framework to model the relabeling-based unlearning method. In particular, we provide a mathematical model for the learning and memorization-based hypotheses. The relabeling-based method is modeled as providing a sequence of *"fabricated evidence"* that conflicts the hypothesis prediction based on existing history of observations.

- Building upon this, we study how the belief in the learning and memorization-based hypotheses change when presented with conflicting evidence by comparing the posterior and prior change relative to the baseline hypotheses that make constant predictions. We show that memorization hypothesis is more stable with an upper bound on its belief change independent of the length of the evidence sequence while the belief change for the learning hypothesis scales with the evidence length.

- We establish a lower bound result showing that even when the unlearn set contains only a single data point, fitting this new set can cause a significant performance gap even if the prior hypothesis space can perfectly fit the original task distribution. This lower bound suggests that in order to fit the new unlearn set while maintaining good overall performance, the model needs to drastically change its prior by expanding the hypothesis space with more complicated hypotheses, which could lead to slower convergence and poor performance.

- Finally, we provide empirical evidence to support our findings by instructing the model to perform binary classification task. We define three types of tasks: LINEAR and RECTANGLE (akin to learning-based generation), where the data shows clear patterns and RANDOM task (akin to memorization-based generation), where the labels are uniformly and randomly generated. Our experiments show that the RANDOM task shows faster and more stable unlearning, maintaining consistently high retain accuracy throughout the unlearning process while the LINEAR and RECTANGLE tasks shows slower convergence and significant fluctuation in retain accuracy. We also include experiments on a dataset with intentionally injected social stereotypes, such as assigning the gender "Female" to all individuals with the profession "nurse," forming a general rule learned by the model. Unlearning the gender attribute in this dataset leads to unstable retain accuracy and slower convergence, compared with a dataset whose gender attribute is randomly assigned and thus closer to the memorization-based setting.

## 1.1 RELATED WORK

**Machine unlearning algorithms for LLMs** The area of machine unlearning for LLMs is rapidly growing with vast amount of literature. This paper focuses on the relabeling-based unlearning method, which has been extensively studied in prior works. For example, (Maini et al., 2024) teach the model to respond with "I don't know" for the prompts in the unlearn set in order to prevent the model from outputting harmful responses. In (Eldan & Russinovich, 2023), the authors replace the unlearn target with its generic counterpart and finetune the model with these alternative labels. Furthermore, (Deeb & Roger, 2024) generates random incorrect choices for multiple-choice questions and optimizes over these choices. Apart from the relabeling-based method, other notable unlearning methods include gradient ascent (Jang et al., 2022; Chen & Yang, 2023), which maximizes the prediction loss of unlearn set, NPO (Zhang et al., 2024; Bronec & Helcl, 2025) which performs preference optimization by treating the unlearn data as negative examples, RMU (Li et al., 2024) which perturbs the activations for the unlearn data while preserving the activations for the retain

data. Other recent LLM unlearning algorithms include (Yao et al., 2024; Liu et al., 2024; Chen & Yang, 2023; Meng et al., 2022; Ishibashi & Shimodaira, 2023) and others.

**Machine unlearning difficulty** Our work is close to the line of work that studies the difficulty of machine unlearning. Previous research has identified several factors that may be related to unlearning difficulty. To name a few, (Krishnan et al., 2025) studies the connection between the frequency of knowledge in the pretrained data and unlearning success. In particular, the authors (Krishnan et al., 2025) find that knowledge with higher frequency is harder to unlearn. (Feng et al., 2025) proposes a Memory Removal Difficulty (MRD) metric to measure the unlearning difficulty for each sample, which can be defined as the stability of data prediction in the presence of model parameter perturbations. Furthermore, (Zhao et al., 2024a) identifies two factors affecting the unlearn difficulty and shows that the unlearning is harder if the sample is more memorized and there is more entanglement between the unlearn and retain data. Finally, (Wu et al., 2025) links the unlearning difficulty to how the knowledge is encoded in the training data and shows that learning with paraphrased descriptions leads to easier unlearning, while unlearning knowledge from a chunk of text is more challenging.

## 2 PRELIMINARIES

### 2.1 LLM UNLEARNING

Large language models (LLMs), parametrized by $\theta$, predict the next word of a sequence $s$ based on a probability distribution $P_\theta(\cdot|s)$. LLM unlearning is the process of removing the undesired data influence of a unlearn dataset, such as private or harmful information without compromising the overall model utility.

To achieve this, we use two distinct datasets. The unlearn dataset ($U$) contains the specific information we want the model to unlearn. The retain dataset ($R$), on the other hand, is a collection of data points that helps the model preserve its original utility and capabilities. By training the model on these two datasets, we can effectively erase the targeted information while keeping the model general utility.

Most existing LLM unlearning methods are achieved by finetuning model $\theta$ over a regularized objective written as

$$\min_\theta (1 - \alpha)L_U(\theta, U) + \alpha L_R(\theta, R)$$

where $L_U$ is the unlearning loss computed on the unlearn set $U$ that measures the unlearning effectiveness, $L_R$ is the retain loss aiming to preserve model utility and $\alpha$ is a weight to balance between the unlearning and model utility maintaining objectives. In previous works, $L_U$ and $L_R$ are implemented in different ways as seen in (Jang et al., 2022; Zhang et al., 2024; Jiang et al., 2020; Li et al., 2024) and others.

### 2.2 RELABELING BASED UNLEARNING

Among the various LLM unlearning methods, an important approach is the relabeling-based method, which has been explored in a sequence of recent work (Yu et al., 2023; Yao et al., 2024; Eldan & Russinovich, 2023; Ishibashi & Shimodaira, 2023; Maini et al., 2024; Deeb & Roger, 2024). The relabeling-based unlearning involves firstly creating a new unlearn set, $U'$, such that each prompt-response pair $(x, y)$ in $U$ is replaced with a modified pair $(x, y')$ where the new response $y'$ is different from the original response $y$. The selection of $y'$ can be either an intentional crafted response, such as 'I don't know' as seen in (Maini et al., 2024) or a randomly selected but sensible response such as a random choice in the context of multiple-choice questions (Deeb & Roger, 2024).

After the construction of $U'$, the overall unlearning objective is

$$\min_\theta (1 - \alpha)L(\theta, U') + \alpha L(\theta, R) \tag{1}$$

where the loss function $L$ is prediction loss. The logic behind relabeling-based unlearning is that by training the model on these relabeled pairs ($U'$), we encourage the model to predict the original prompts with alternating or neutral responses, effectively overwriting the undesired information.

Some prior works combine the relabeling-based objective (1) with other unlearning techniques, such as gradient ascent loss, to form a more comprehensive optimization objective. This work will focus exclusively on the objective defined in (1) given that it remains a fundamental component in these works and focusing on equation (1) can obtain deeper insight into how relabeling works without the influence from other unlearning techniques.

## 3 MODEL'S BELIEFS UPDATE FOR UNLEARNING VIA RELABELING

In this section, we model the relabeling-based unlearning method as presenting conflicting evidence to the model (referred to as *fabricated evidence*). We study how this new evidence updates the model's internal beliefs, especially for the learning and memorization-based hypothesis, whose detailed definitions will be provided later in this section.

Generally speaking, a learning-based hypothesis is a general rule learned by the model to map the output to input (e.g. the arithmetic rule to give the response to '*given the equation 3x+5=11, the solution for x is 2*'). In contrast, a memorization-based hypothesis involves memorizing a specific output for each input, usually for arbitrary information (e.g. '*The file name of Alice's medical record is 187465373622*').

After initial training, the model holds a certain belief in these hypotheses on how the response is generated based on the prompts. The relabeling-based method, as defined in equation (1), can be seen as presenting the evidence to the model that contradicts the model's current beliefs. For example, training with the pair '*given the equation 3x+5=11, the solution for x is 3*' challenges the arithmetic rule the model uses for basic calculation. Similarly, training with the pair '*the file name of Alice's medical record is 17384748343*' contradicts specific information the model memorizes previously. The process of the relabeling-based method makes the model update its belief of the hypotheses in the presence of conflicting evidence.

We show that learning-based hypothesis is less stable than memorization-based hypothesis when faced with fabricated evidence that contradicts the hypothesis's prediction. When a learning-based hypothesis is presented with fabricated conflicting evidence, its belief relative to baseline hypotheses decreases exponentially with the length of the evidence. In contrast, a memorization-based hypothesis is relatively stable. Its belief relative to baseline hypotheses only drops by a constant amount regardless of the length of the fabricated evidence, as long as the same input has only been observed by the model a limited number of times. We also show that the learning-based hypothesis when confronted with fabricated evidence requires the model to search for a new hypothesis that fits both the modified unlearn and retain data, which can be a slow process when the model needs to drastically change its underlying priors on the hypothesis space. In contrast, the memorization-based hypothesis is more flexible and efficient to make alternative predictions.

### 3.1 BELIEF UPDATE MODELING

Given a sequence of prompt-response pairs $D = \{(x_j, z_j)\}_{j=1}^n$, we simplify our analysis by letting $x_j \in [N] = \{0, 1, 2, \ldots N\}$ and $z_j$ is either $-1$ or $+1$ for any $j \in [n]$. Assume that the prompt $x$ is uniformly sampled from $[N]$ and the observed response $z \in \{-1, +1\}$ is a noisy version of the true, underlying response $y \in \{-1, +1\}$. Specifically, the true response $y$ is flipped with a probability $\epsilon \in (0, 0.5)$ independently for each observation.

We propose two primary hypotheses to distinguish between how the model predicts the response $y$ given input $x$, a learning-based hypothesis ($H_0$) that the model learns the underlying, general rule between $x$ and $y$, and a memorization-based hypothesis ($H_1$) that the model memorizes every pair it has ever seen. The mathematical representations of both hypotheses are defined as follows:

- $H_0$ (Learning-based Hypothesis): The response $y$ is determined by a known function $f : [N] \to \{-1, +1\}$, that is, $y = f(x)$.

- $H_1$ (Memorization-based Hypothesis): A latent vector $V \in \{-1, +1\}^{N+1}$ is sampled once where each element in $V$ is i.i.d uniformly sampled from $\{-1, +1\}$. The response $y$ for input $x$ is the value stored at the $x_{th}$ position of vector $V$, i.e., $y = V_x$.

Under the learning-based hypothesis ($H_0$), the model learns a general rule $f(\cdot)$ that characterizes the relationship between the prompt $x$ and response $y$. In contrast, under the memorization-based hypothesis ($H_1$), the model acts like a lookup table or a database, and the relationship between the input $x$ and output $y$ is completely arbitrary and random, determined by the initial sampling of the latent vector $V$. Also, since each element in $V$ is sampled independently, there is no dependency between the responses for different inputs. The response of input $x_1$ provides no information for the response of input $x_2$, which means there is no underlying, general rule for the model to learn. The optimal strategy is to memorize the responses for each input individually.

One key distinction between learning and memorization-based hypotheses lies in their uncertainty on unobserved examples. Memorization-based generation assumes independence between inputs, leading to high uncertainty on unobserved examples. In contrast, learning-based generation generalizes rules across inputs, resulting in high confidence and low uncertainty for predictions on unseen data.

We compare the prior-posterior belief change of a hypothesis relative to baseline hypotheses. We define two baseline hypotheses that make constant predictions:

- $H_2$: $y = +1$ for all $x \in [N]$.
- $H_3$: $y = -1$ for all $x \in [N]$.

**Notations**    we denote $h$ as the history of $n$ observations, and the prior belief of a hypothesis $H$ given observations $h$ as $P(H|h)$. New evidence is denoted as $e$, which consists of $k$ observations. The posterior belief of $H$ after incorporating new evidence $e$ is denoted as $P(H|h, e)$. Finally, $P(y|H, h, x)$ represents the prediction probability of the label $y$ for input $x$ given that hypothesis $H$ is true and a history of observations $h$.

We also define fabricated evidence against a hypothesis as a sequence of data points that consistently contradict the hypothesis's predictions, given its past observations.

**Definition 1.** (Fabricated Evidence) Given a hypothesis $H$ with history observations $h$, we call a sequence of datapoints $e = \{(x_j, z_j)\}_{j=1}^k$ fabricated evidence against $H$, if we have

$$y_j = - \arg\max_{y \in \{-1, +1\}} P(y|H, h, x_j) \quad \forall j \in [k]$$

and $z_j$ is a noisy observation of $y_j$ with i.i.d flipping noise $\epsilon$.

We will first test the learning-based hypothesis $H_0$ against the baseline hypotheses $H_2$ and $H_3$. Our primary focus will be to study the stability of the belief of $H_0$ in the presence of fabricated evidence $e$. To simplify the analysis, we consider a specific case where all evidence shares the same input $x^*$. In particular, Theorem 2 shows that the logarithm of the belief drop of the learning hypothesis $H_0$ relative to the baseline hypothesis scales linearly with the evidence length $k$ with high probability, whose proof can be found in Appendix A.

**Theorem 2.** *(Stability of Learning-based Hypothesis) Let $P(H_0|h)$, $P(H_2|h)$ and $P(H_3|h)$ be existing belief priors based on a history of observations $h$. Consider fabricated evidence $e = \{(x_j, z_j)\}_{j=1}^k$ against $H_0$ with history $h$, where $x_j = x^*$ for all $j \in [k]$. There exists an $i \in \{2, 3\}$ such that the change on the log-posterior is given as*

$$\Delta_e = \log\left(\frac{P(H_0|h)}{P(H_i|h)}\right) - \log\left(\frac{P(H_0|h, e)}{P(H_i|h, e)}\right) = (k - 2l)\log\left(\frac{1-\epsilon}{\epsilon}\right)$$

*where $l$ is the number of flipped observations in the evidence $e$, which follows a binomial distribution $l \sim Binomial(k, \epsilon)$. Furthermore, since $\epsilon \in (0, 0.5)$, we have with probability over $1 - O(k^{-10})$,*

$$\Delta_e = \Omega(k)$$

*where the randomness is taken over the observation noise in the evidence $e$.*

**Remark:**    Theorem 2 shows that when presented a sequence of fabricated evidence against $H_0$, the log-posterior of the learning-based hypothesis ($H_0$) relative to the baseline hypothesis ($H_2$ or $H_3$) decrease linearly to the length of the evidence. When the evidence length is sufficiently long, the belief of $H_0$ will be overwhelmed by the baseline hypothesis. Note that this result holds regardless

of the choice of $x^*$ for which the evidence is collected. Even if $x^*$ was not observed in the initial history $h$, fabricated evidence regarding $x^*$ still causes a belief drop for $H_0$.

Meanwhile, in the next theorem (Theorem 3), we will show that the memorization-based hypothesis ($H_1$) is relatively stable, and may lead to constant belief drop regardless of the length of the fabricated evidence. The proof of Theorem 3 can be found in Appendix A.

**Theorem 3.** *(Stability of Memorization-based Hypothesis) Let $P(H_1|h)$, $P(H_2|h)$ and $P(H_3|h)$ be existing belief priors based on a history of observations $h$. Consider fabricated evidence $e = \{(x_j, z_j)\}_{j=1}^k$ against $H_1$ with history $h$, where $x_j = x^*$ for all $j \in [k]$.*

*Let $h_{x_*}$ be the subset of the history $h$ with input value $x^*$, $m_{1,x^*}$ be the number of $z_j = 1$ in $h_{x^*}$ and $m_{-1,x^*}$ be the number of $z_j = -1$ in $h_{x^*}$, then we have for any $i \in \{2, 3\}$, the change on the log-posterior can be given by*

$$\Delta_e = \log\left(\frac{P(H_1|h)}{P(H_i|h)}\right) - \log\left(\frac{P(H_1|h,e)}{P(H_i|h,e)}\right) \leq \log\left(1 + \left(\frac{1-\epsilon}{\epsilon}\right)^{|m_{1,x^*} - m_{-1,x^*}|}\right)$$

*In particular, if $x^*$ is not observed in the initial history $h$, then we have the belief update*

$$\Delta_e \leq \log(2)$$

**Remark:** Theorem 3 states that the belief drop for memorization-based hypothesis $H_1$ relative to the baseline hypotheses ($H_2$ and $H_3$) is upper bounded by a value that doesn't depend on the length of the evidence $k$. This means that even as the evidence length approaches infinity, the belief drop is limited. Moreover, unlike the learning case where the belief drop is independent of the chosen input $x^*$, in the memorization case, the belief update varies based on the choice of $x^*$. Specifically, the more frequently $x^*$ has appeared in the initial history $h$ and more consistent its observations are, the greater the belief drop it will cause when there is fabricated evidence against it.

### 3.2 Prediction Update for Relabeling-based Unlearning

We also note that the learning and memorization-based hypotheses have different ways for the response predictions. For the learning-based hypothesis $H_0$, the entire prediction behavior is governed by the prediction function $f$. Therefore, in order to minimize both the unlearn loss $L(\theta, U')$ and the retain loss $L(\theta, R)$, the model has to find a new hypothesis $H'$ that better fits the datasets $U'$ and $R$. However, how fast the model can find this new hypothesis $H'$ highly depends on the model's prior, like what the hypothesis could be or what hypothesis space $H'$ belongs to. In the next theorem, we show that given a prior hypothesis class $\mathcal{H}$ that can perfectly fit the task distribution, then any hypothesis $h \in \mathcal{H}$ that achieves good accuracy on $U'$ will suffer from big performance drop even in the case that $|U'| = 1$. This lower bound, whose proof is given in Appendix A, is stated as follows:

**Theorem 4.** *Given a $d$-dimensional linear hypothese class defined as $\mathcal{H} = \{h_{w,b}(x) = sign(w^T x + b)|w \in \mathbb{R}^d, b \in \mathbb{R}\}$. There exists a distribution $\mathcal{D}$ and $U' = \{(x', y') \in [N]^d \times \{-1, +1\}\}$ where with $N \geq 3$, such that $\min_{h \in \mathcal{H}} err_D(h) = 0$ and $P_{\mathcal{D}}(x = x')$ is negligible, however, for any $\hat{h} \in \mathcal{H}$ with $\hat{h}(x') = y'$, we have*

$$err_D(\hat{h}) \geq 0.1$$

*where $err_{\mathcal{D}}(\cdot)$ is the 0-1 error evaluated on distribution $\mathcal{D}$.*

The theorem above states a fundamental challenge for models that rely on a learning-based hypothesis when using the relabeling-based unlearning method. In order to minimize both the unlearning and retain loss, the model sometimes needs to drastically change its prior beliefs about the functions required to fit both $U'$ and the retain set $R$. In the context of Theorem 4, even though the original task can be perfectly predicted by a linear classifier, fitting the modified unlearn set $U'$ (let $\hat{h}(x') = y'$) using linear classifiers will lead to a constant error rate. As a result, the model needs to expand the hypothesis class to include more complex hypotheses. This change in prior beliefs can make the optimization process slower and more complex, which we will provide empirical evidence in the experiments presented in the next section.

In contrast, the memorization hypothesis ($H_1$) gives a more flexible way to predict the response. In particular, $H_1$ admits a decision rule based on maximum likelihood estimation written as

$$\hat{y}_x = \begin{cases} +1 & \text{if } m_{1,x} \geq m_{-1,x} \\ -1 & \text{if } m_{1,x} < m_{-1,x} \end{cases}$$

where $m_{1,x}$ and $m_{-1,x}$ are the number of times the model has observed prompt $x$ with response $+1$ or $-1$ respectively. Therefore, in order to change the model prediction form $+1$ to $-1$ for specific prompt $x$, we can repeatedly present the model with observations of $(x, y' = -1)$. Eventually, $m_{-1,x}$ will be larger than $m_{1,x}$, at which point, the model's prediction for $x$ is flipped to $y' = -1$ and the information associated with $x$ is effectively unlearned. This process is restricted to specific prompt and does not require a search for new hypotheses as in the learning case. We will show empirically in the next section that such restricted and localized update leads to faster and more stable unlearning.

## 4 EXPERIMENTS

In this section, we present our experimental results, which empirically validate the theoretical claims from the previous section that relabeling-based unlearning is more effective for memorization-based generation than for learning-based generation. We begin with experiments in which we instruct LLMs to perform a binary classification task, consistent with our theoretical setting. We then provide additional experiments on unlearning social stereotypes, a setting more closely aligned with practical LLM applications. These results are deferred to Appendix B.4 due to space constraints.

### 4.1 EXPERIMENT SETUP

In our experiments, we follow prior work (Zhao et al., 2024b; Dinh et al., 2022) to instruct the model to perform binary classification task. The model is given a two-dimensional input $x = (x_1, x_2)$ where $x_1$ and $x_2$ are integers between 0 and 200. The prediction label is either $-1$ and $+1$ which are mapped to class name "Foo" and "Bar", respectively. We choose this binary classification task for the following key reasons: 1) the nature of the task makes it straightforward to construct specific data distribution patterns and conduct experiments under a well-controlled setup. 2) it enables a clear visualization of the model's decision boundary, which allows us to observe how the model's beliefs update concerning the underlying data generation hypothesis.

An example of the prompts used in our experiments is shown as follows:

What is the label for this input?\n Input: 62 87\n Label: Foo

We generate prompts of this form for each $(x, y)$ pair in the dataset.

**Tasks and Datasets**  We define three different tasks:

- *LINEAR*: data points $(x, y)$ are linearly separable with $y = \text{sign}(x_1 - x_2)$.
- *RECTANGLE*: the input domain is partitioned into 4 quadrants centered at $(100, 100)$. Points in two quadrants are labeled as $+1$ and those in the remaining quadrants are labeled as $-1$.
- *RANDOM*: label $y$ is uniformly and randomly generated from $\{-1, +1\}$ for each input $x$.

For each task, we generate 1024 datapoints in each task and choose 30 of them as the unlearn set $U$, while the rest will be the retain set $R$. The data distribution of all tasks are plotted in Figure 1.

**Language model**  The language model we use is Llama-3.2-3B-Instruct (Dubey et al., 2024). Similar results are obtained with other models including Qwen3-4B (QwenTeam, 2025) and Llama-3.2-8B-Instruct (Dubey et al., 2024)), which are deferred to Appendix B.2.

**Training/Unlearning method**  We first finetune the original model on the entire dataset for each task. After finetuning, we perform unlearning on the unlearn set using the *relabeling-based method* defined in equation (1). In particular, we construct a modified unlearn set $U'$ by flipping the label for each data point in $U$ to generate alternative responses and then train the model to fit both $U'$ and

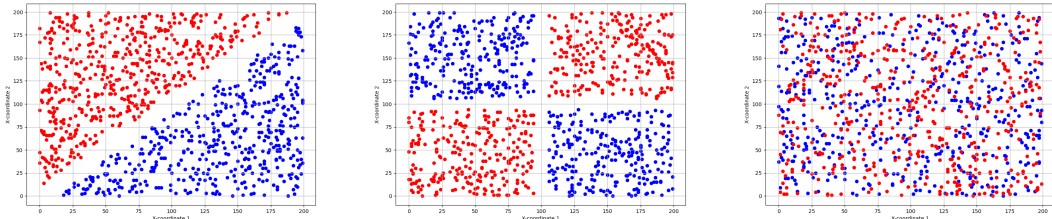

Figure 1: Data visualization for the LINEAR (left), RECTANGLE (middle) and RANDOM (right) tasks.

the retain set $R$. The prediction loss $\ell(\theta, (x, y)) = -\log(P_\theta(y|x))$ is used as the training objective for both fine-tuning and unlearning. The loss is calculated only on the label $y$. More detailed hyperparameter settings are provided in Appendix B.1.

## 4.2 RESULTS

**Decision Boundary for Learning and Memorization-based Generations** Since it is challenging to directly obtain the underlying model hypothesis of the model, we instead plot the decision boundary as an indirect way to understand the model's prediction process. The decision boundaries for each task after the initial finetuning are provided in Figure 2.

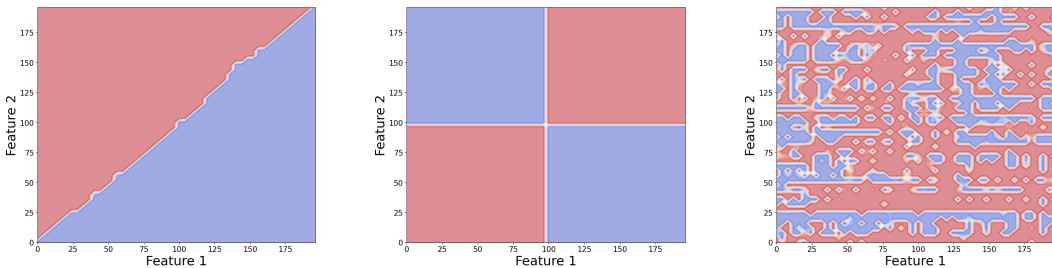

Figure 2: Decision boundaries for the LINEAR task (left), RECTANGLE (middle) and the RANDOM task (right) after finetuning. The clear, regular decision boundaries for the LINEAR and RECTANGLE tasks demonstrate that the model learned the underlying data generation rule, indicating the model's generation is more learning-based. In contrast, the irregular and scattered decision boundary of the RANDOM task suggests the prediction relies more on memorization rather than rule learning.

From Figure 2, we can see a clear discrepancy between tasks relying on learning-based generation and memorization-based generation. The decision boundaries for the LINEAR and RECTANGLE tasks (left and middle of Figure 2) show clear and regular patterns. This suggests that the model successfully learns the underlying rules of data generation, showing a **learning-based generation** approach. In contrast, the model finetuned on the RANDOM task data is akin to **memorization-based generation** as the labels are uniformly and randomly generated and there is no underlying data distribution structure for the model to learn. As a result, its decision boundary (right of Figure 2) is irregular and scattered, showing no clear pattern.

**Unlearn Efficiency** Here, we show that unlearning is more efficient for the memorization-based task (RANDOM) than for the learning-based tasks (LINEAR and RECTANGLE). The unlearning performance is evaluated using the accuracy on the retain set $R$ (retain accuracy) and the accuracy on the unlearn set $U$ (unlearn accuracy), as shown in Figure 3.

We observe relabeling-based method achieves nearly $100\%$ retain accuracy and zero unlearn accuracy for all tasks by the end of unlearning. However, the unlearning process shows different patterns between learning and memorization-based tasks.

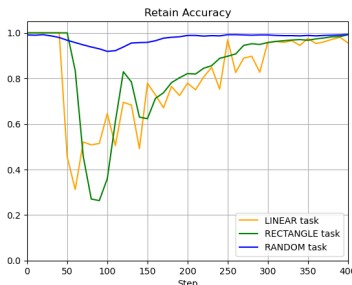 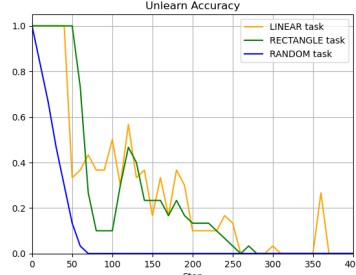

Figure 3: Accuracy for the retain set $R$ (left) and unlearn set $U$ (right) during unlearning for different tasks. The unlearn set consists of 30 data records, which represent 3% of the full dataset. Unlearning the RANDOM task shows faster and more stable convergence compared to the learning-based tasks (LINEAR and RECTANGLE). Retain accuracy remains consistently high (over 90%) for the RANDOM task but drops sharply to under 40% for the other two tasks in the middle of the unlearning before recovering. Meanwhile the unlearn accuracy for the RANDOM task reaches zero faster than those for the other two tasks.

- (Unlearning Stability) The stability of the unlearning is reflected by the retain accuracy during unlearning (left of Figure 3). A stable unlearning process can effectively remove information from the unlearn set without significantly affecting the retain accuracy. The retain accuracy for the LINEAR and RECTANGLE task drops sharply under 40% during unlearning before recovering. In contrast, the retain accuracy of the RANDOM task remains above 90% consistently. This indicates greater stability of the memorization-based task.

- (Unlearning Rate) The rate of unlearning is captured by the unlearn accuracy (right of Figure 3). A faster unlearn rate means the accuracy on unlearn set $U$ drops more quickly. The unlearn accuracy of the RANDOM task decreases significantly faster, reaching zero accuracy in around 70 steps, while the LINEAR and RECTANGLE tasks require over 250 steps to achieve the same level. The slower convergence can be a result of the requirement of the learning-based tasks to search for new hypotheses to simultaneously fit the modified unlearn set $U'$ and retain set $R$, as discussed in section 3.2.

**Decision Boundary Evolution during Unlearning**    To observe the change of the underlying hypotheses the model employs for the prediction, we save snapshots of model at various steps of the unlearning and plot the decision boundary for each saved model. In Figure 4, we show how the decision boundary evolves at different stages of the unlearning.

The top and middle row in Figure 4 plots the decision boundary evolution for learning-based tasks (LINEAR and RECTANGLE). Despite the unlearn set representing only 3% of the full dataset, the model's belief in the original learning-based hypothesis gets significantly shattered at the beginning of the unlearning due to the fabricated conflicting evidence introduced by $U'$, leading to a drastic change in the model's prediction behavior with an unclear and irregular decision boundary. The model then gradually refines its internal belief and successfully finds a new hypothesis that can fit both $U'$ and $R$ by the end of the unlearning.

Meanwhile, the bottom row of Figure 4 illustrates the decision boundary evolution during unlearning for the RANDOM task. Even though some localized changes on the decision boundary are still visible, there are no significant global changes and the overall decision boundary structure remains stable, which indicates the stability of the model belief of the memorization case for the relabeling-based unlearning method.

## 5    CONCLUSION

In this paper, we investigate how the nature of a model's generation influences the efficiency of relabeling-based unlearning, with a focus on distinguishing between learning-based and memorization-based generations. Our results show that relabeling-based methods are more effective for unlearning memorization-based generation, exhibiting more stable belief updates and requiring

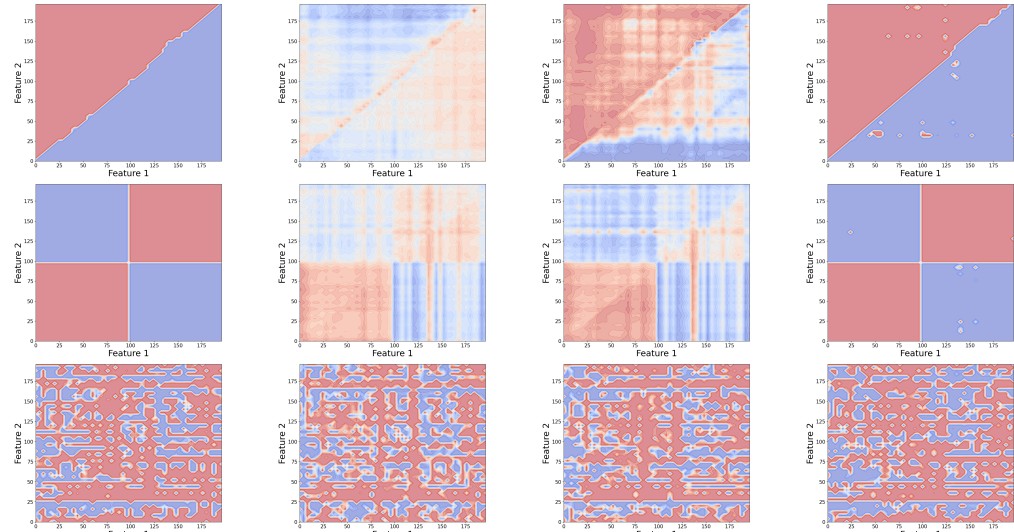

Figure 4: Decision boundary evolution during unlearning for tasks: LINEAR (top row), RECT-ANGLE (middle row) and RANDOM (bottom row). The figures show decision boundaries at unlearning steps: 0, 60, 180, 420. The unlearn set consists of 30 data records, which represent 3% of the full dataset. Light-colored areas indicate higher model uncertainty. The learning-based tasks (LINEAR and RECTANGLE) show significant, global changes on their decision boundary as the model unlearns, suggesting significant shift on the model's belief. In contrast, memorization-based task (RANDOM) only shows localized, minor boundary updates, highlighting the stability of the model's belief.

no significant changes to the model's priors. In contrast, unlearning learning-based generation is inherently more challenging. Promising future directions include extending our theoretical framework to settings with more general types of response other than binary classification, demonstrating our results on a broader range of benchmarks, and exploring connections to LLM alignment, which shares conceptual similarities with relabeling-based unlearning.

## REPRODUCIBILITY STATEMENT

All datasets used in our experiments, along with the experiment code, are provided as supplementary material. The package also includes a README file with detailed instructions for reproducing the experiments in this paper. Proofs of the theoretical results can be found in Appendix A.

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

## A   MISSING PROOF IN SECTION 3

**Theorem** (Restatement of Theorem 2). *Let $P(H_0|h)$, $P(H_2|h)$ and $P(H_3|h)$ be existing priors based on a history of observations $h$. based on a history of observations $h$. Consider fabricated evidence $e = \{(x_j, z_j)\}_{j=1}^{k}$ against $H_0$, where $x_j = x^*$ for all $j \in [k]$. Then there exists an $i \in \{2, 3\}$ such that the change on the log-posterior is given as*

$$\Delta_e = \log\left(\frac{P(H_0|h)}{P(H_i|h)}\right) - \log\left(\frac{P(H_0|h, e)}{P(H_i|h, e)}\right) = (k - 2l)\log\left(\frac{1 - \epsilon}{\epsilon}\right)$$

*where $l$ is the number of flipped observations in the evidence $e$, which follows a binomial distribution $l \sim Binomial(k, \epsilon)$. Furthermore, since $\epsilon \in (0, 0.5)$, we have with probability over $1 - O(k^{-10})$,*

$$\Delta_e = \Omega(k)$$

*where the randomness takes from the observation noise in $e$.*

*Proof.* Without loss of generality, we assume that $f(x^*) > 0$, then $y_i = -1$ for all $y_i$ from the new evidence $e$. Then by Bayes' theorem, we have

$$\Delta_e = \log\left(\frac{P(H_0|h)}{P(H_3|h)}\right) - \log\left(\frac{P(H_0|h, e)}{P(H_3|h, e)}\right) = \log\left(\frac{P(e|H_3, h)}{P(e|H_0, h)}\right)$$

Next, we focus on the ratio $\frac{P(e|H_3, h)}{P(e|H_0, h)}$. First, since both $H_0$ and $H_3$ assume i.i.d generation, therefore $P(e|H_0, h) = P(e|H_0)$ and $P(e|H_3, h) = P(e|H_3)$. Then given the evidence $e = (x^*, z_1)\ldots(x^*, z_k)$, let $l$ be the number of $+1$ in all $z_i$s, then we have

$$P(e|H_0) = \left(\frac{1}{N+1}\right)^k (1 - \epsilon)^l \epsilon^{k-l}$$

and

$$P(e|H_3) = \left(\frac{1}{N+1}\right)^k \epsilon^l (1 - \epsilon)^{k-l}$$

Combining everything together, we have

$$\Delta_e = \log\left(\frac{P(e|H_3, h)}{P(e|H_0, h)}\right) = (k - 2l)\log\left(\frac{1 - \epsilon}{\epsilon}\right)$$

Since $y_i = -1$ in $e$, $l$ also denotes the total number of times $z_i$ flip $y_i$ which happens independently with probability $\epsilon$. Therefore, $l$ is binomial distributed with $l \sim Binomial(k, \epsilon)$. Since $\epsilon < 0.5$, via McDiarmid's inequality, we have with probability over $1 - O(k^{-10})$

$$k - 2l = \Omega(k)$$

where the randomness takes from the noise on the observation $z$.

Similar proof can be obtained for $\Delta_e$ between $H_0$ and $H_2$, when $f(x^*) \leq 0$.

$\square$

**Theorem** (Restatement of Theorem 3). *Let $P(H_1|h)$, $P(H_2|h)$ and $P(H_3|h)$ be existing priors based on a history of observations $h$. Consider fabricated evidence $e = \{(x_j, z_j)\}_{j=1}^{k}$ against $H_1$ with history observations $h$, where $x_j = x^*$ for all $j \in [k]$.*

*Let $h_{x_*}$ be the subset of the history $h$ with input value $x^*$. Let $m_{1, x^*}$ be the number of $z_j = 1$ in $h_{x^*}$ and $m_{-1, x^*}$ be the number of $z_j = -1$ in $h_{x^*}$, then we have for any $i \in \{2, 3\}$, the change on the log-posterior can be given by*

$$\Delta_e = \log\left(\frac{P(H_1|h)}{P(H_i|h)}\right) - \log\left(\frac{P(H_1|h, e)}{P(H_i|h, e)}\right) \leq \log\left(1 + \left(\frac{1 - \epsilon}{\epsilon}\right)^{|m_{1, x^*} - m_{-1, x^*}|}\right)$$

*In particular, if $x^*$ is not observed in the initial history $h$, then we have the belief update*

$$\Delta_e \leq \log(2)$$

*Proof.* Without loss of generality, we assume that the label $y_i = 1$ for $i \in [k]$ in evidence $e$, Then we are comparing the belief between $H_1$ and $H_3$ and $H_2$.

In particular, we have for $i \in \{2, 3\}$

$$\Delta_e = \log\left(\frac{P(H_1|h)}{P(H_i|h)}\right) - \log\left(\frac{P(H_1|h, e)}{P(H_3|h, e)}\right) = \log\left(\frac{P(e|H_i, h)}{P(e|H_1, h)}\right) \tag{2}$$

Next, we need to compute $P(e|H_1, h)$. Denote random variable $y_{x^*}$ as the corresponding $y$ to $x^*$. Then we have

$$P(e|H_1, h) = \sum_{y \in \{-1,1\}} P(e|y_{x^*} = y, H_1, h) \cdot P(y_{x^*} = y|H_1, h)$$

Then we define the $l$ be the number of times $z_i = 1$ in $e$, it is straightforward to obtain that

$$P(e|y_{x^*} = -1, H_1) = \left(\frac{1}{N+1}\right)^k \epsilon^l (1-\epsilon)^{k-l}$$

$$P(e|y_{x^*} = 1, H_1) = \left(\frac{1}{N+1}\right)^k \epsilon^{k-l} (1-\epsilon)^l$$

Meanwhile, we have

$$P(e|H_2, h) = \left(\frac{1}{N+1}\right)^k \epsilon^{k-l} (1-\epsilon)^l$$

$$P(e|H_3, h) = \left(\frac{1}{N+1}\right)^k \epsilon^l (1-\epsilon)^{k-l}$$

Next, we will compute the

$$P(y_{x^*} = y|h, H_1) \quad \text{for } y \in \{-1, 1\}$$

We denote $h_{x^*}$ the subset of $h$ that have the input value $x^*$, it is easy to show that

$$P(y_{x^*} = y|h, H_1) = P(y_{x^*} = y|h_{x^*}, H_1) \quad \text{for } y \in \{-1, 1\}$$

This is due to the fact that under $H_1$, the part of $h$ whose feature is not $x^*$ is independent of $y_{x^*}$.

Applying Bayes' theorem, we obtain

$$P(y_{x^*} = 1|h_{x^*}, H_1) = \frac{P(h_{x^*}|y_{x^*} = 1, H_1) \cdot P(y_{x^*=1}|H_1)}{P(h_{x^*}|H_1)}$$

It is easy to show that

$$P(y_{x^*} = 1|h_{x^*}, H_1) = \frac{(1-\epsilon)^{m_1} \epsilon^{m_{-1}}}{(1-\epsilon)^{m_1} \epsilon^{m_{-1}} + (1-\epsilon)^{m_{-1}} \epsilon^{m_1}}$$

Similarly, we have

$$P(y_{x^*} = -1|h_{x^*}, H_1) = \frac{(1-\epsilon)^{m_{-1}} \epsilon^{m_1}}{(1-\epsilon)^{m_1} \epsilon^{m_{-1}} + (1-\epsilon)^{m_{-1}} \epsilon^{m_1}}$$

Then we have

$$\frac{P(e|H_2, h)}{P(e|H_1, h)} = \frac{P(e|H_2, h)}{\sum_{y \in \{-1,1\}} P(e|y_{x^*} = y, H_1) \cdot P(y_{x^*} = y|H_1, h)}$$

$$\leq \frac{P(e|H_2, h)}{P(e|y_{x^*} = 1, H_1) \cdot P(y_{x^*} = 1|H_1, h_{x^*})}$$

$$= \frac{1}{P(y_{x^*} = 1|H_1, h_{x^*})} \leq 1 + \left(\frac{1-\epsilon}{\epsilon}\right)^{|m_{1,x^*} - m_{-1,x^*}|}$$

The last equality follows that

$$P(e|H_2, h) = P(e|y_{x^*} = 1|H_1, H_1) = \left(\frac{1}{N+1}\right)^k \epsilon^{k-l}(1-\epsilon)^l$$

Similarly, we also have

$$\frac{P(e|H_3, h)}{P(e|H_1, h)} \leq 1 + \left(\frac{1-\epsilon}{\epsilon}\right)^{|m_{1,x^*} - m_{-1,x^*}|}$$

Combining these results with equation equation 2, we obtain the desired results.

$\square$

**Theorem** (Restatement of Theorem 4). *Given a d-dimensional linear hypotheses class defined as* $\mathcal{H} = \{h_{w,b}(x) = sign(w^T x + b)|w \in \mathbb{R}^d, b \in \mathbb{R}\}$. *There exists a distribution* $\mathcal{D}$ *and a data record* $z' = (x, y') \in [N]^d \times \{-1, +1\}$ *where* $[N] = \{1, 2, 3, \ldots N\}$ *with* $N \geq 3$, *such that* $\min_{h \in H} err_D(h) = 0$ *and* $P_D(x = x')$ *is negligible, however, for any* $\hat{h} \in \mathcal{H}$ *such that* $\hat{h}(x) = y'$, *we have*

$$err_D(\hat{h}) \geq 0.1$$

*where* $err_D(\cdot)$ *is the 0-1 error evaluated on distribution D.*

*Proof.* The construction is as follows:

- The points are uniformly distributed in hyper cube $C = \{1, 2, 3\}^d$ where $P(x \in C) = 0.2$

- The ground truth for all points in $x$ is 1, that is, $P(y = 1|x \in C) = 1$.

The unlearn point is the center of the hypercube $x' = \{2\}^d$. The rest of distribution can be constructed such that, $\exists h^* \in \mathcal{H}$, such that $err_D(h^*) = 0$, for example with arbitrary distribution for $x \notin C$ and $P(y = 1) = 1$ for all $x$ with $x_1 < 6$ and $P(y = -1) = 1$ for all $x_1 \geq 6$ where $x_1$ is the first coordinate of $x$.

Now, we show that if we want to "unlearn" $x'$ by finding a $\hat{h} \in \mathcal{H}$ with parameters $w, b$ such that $h_{w,b}(x') \neq 1$, we have

$$err_D(\hat{h}) \geq 0.1$$

First, since $\hat{h}(x') = -1$, then we have $w^T x + b \leq 0$. Now we consider other points in the unit $B$, since B is centered as $x'$, then any point $x$ can be written

$$x = x' + \Delta x$$

where $\|\Delta x\| \leq 1$. Note that since $x$ is uniformly distributed across $B$, which implies that the direction of $\Delta x$ is symmetric, that is, for any $x = x' + \Delta x \in B$, there exits an $\bar{x} = x' - \Delta x$ such that $\bar{x} \in B$ and $P_D(x) = P_D(\bar{x})$. Therefore, we have

$$P_D(w^T \Delta x \leq 0|x \in B) \geq 0.5$$

For any $x$ with $w^T \Delta x \leq 0$, we have

$$w^T x + b = w^T(x' + \Delta x) + b = w^T x' + w^T \Delta x + b \leq 0$$

That implies that at least these $x$ will also be classified as $-1$, which is

$$P_D(\hat{h}(x) = -1|x \in B) \geq 0.5$$

Combined with fact that $P_D(x \in B) \geq 0.2$ and the true label of $x \in B$ is $+1$, we have

$$err_D(\hat{h}) \geq 0.2 \times 0.5 = 0.1$$

Since we consider realizable case, we also have

$$err_D(\hat{h}) - \min_{h \in \mathcal{H}} err_D(h) \geq 0.1$$

$\square$

## B EXPERIMENT DETAILS

### B.1 EXPERIMENT SETUP

We run our experiments with Llama3.2-3B-Instruct, Qwen3-4B and Llama3.2-8B-Instruct models. We use Lion optimizer (Chen et al., 2023) for both finetuning and unlearning with a learning rate of $5 \times 10^{-7}$ and a batch size of 8 for all experiments. No learning rate scheduler is used. We train 20 epochs for finetuning. The regularization parameter $\alpha$ in equation (1) is set to 0.5 across all experiments. A gradient norm clipping equal to 1 is added for unlearning experiments. All experiments are conducted on a single NVIDIA A100 GPU.

### B.2 ADDITIONAL EXPERIMENTS WITH OTHER MODELS

**Overview** We provide additional experimental results for finetuning and unlearning with Qwen3-4B and Llama3.2-8B-Instruct. The results for Qwen3-4B are shown in Figures 5-7 and those for Llama3.2-8B-Instruct are provided in 8-10. In particular, the decision boundaries after initial fine-tuning are in Figure 5 for Qwen3-4B and Figure 8 for Llama3.2-8B-Instruct. The retain and unlearn accuracies during unlearning are in Figure 6 for Qwen3-4B and Figure 9 for Llama3.2-8B-Instruct. Finally, the decision boundary evolutions for each task are presented in Figure 7 for Qwen3-4B and Figure 10 for Llama3.2-8B-Instruct.

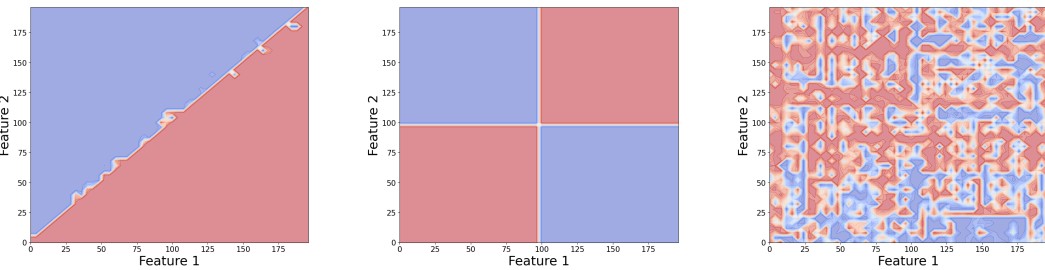

Figure 5: Decision boundary for the LINEAR task (left), RECTANGLE (middle) and the RANDOM task (right) after finetuned on Qwen3-4B.

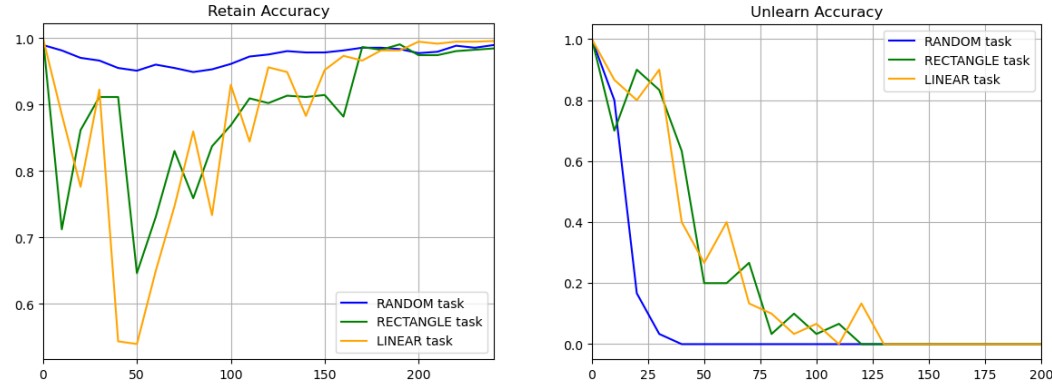

Figure 6: Retain and unlearn accuracies for the LINEAR (left), RECTANGLE (middle) and RANDOM (right) task for Qwen3-4B model.

**Discussion** The experiments for the Qwen3-4B and Llama3.2-8B-Instruct show results consistent with those reported in section 4 for Llama3.2-3B-Instruct. In particular, all models successfully learn the underlying data generation rule for LINEAR and RECTANGLE tasks, leading to clear and regular decision boundaries. The decision boundaries for RANDOM task remain scattered and irregular for all models. This indicates that similar model behaviors occur after finetuning for models with different sizes and architectures.

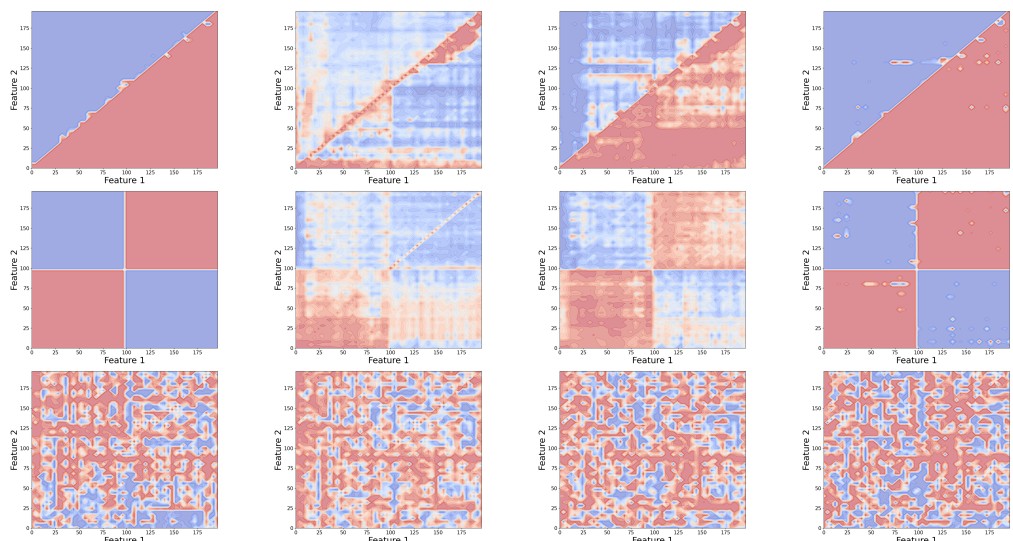

Figure 7: Decision boundary evolutions during unlearning for LINEAR (top row), RECTANGLE (middle row) and RANDOM (bottom row) tasks for Qwen3-4B model. These figures show decision boundaries at unlearning steps: 0, 50, 100, 200.

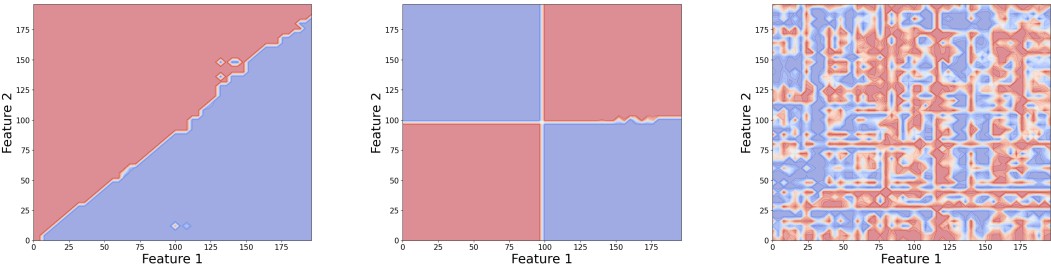

Figure 8: Decision boundary for the LINEAR task (left), RECTANGLE (middle) and the RANDOM task (right) after finetuning for Llama3.2-8B-Instruct.

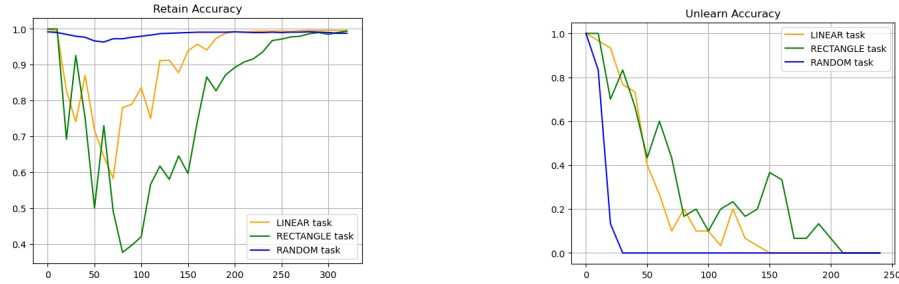

Figure 9: Accuracy for the retain set $R$ (left) and unlearn set $U$ (right) during unlearning for different tasks for Llama3.2-8B-Instruct.

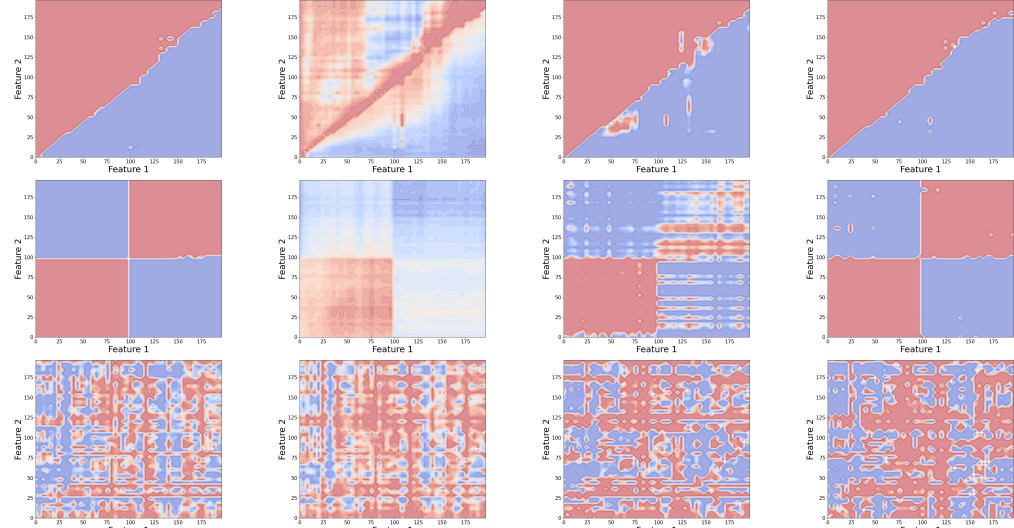

Figure 10: Decision boundary evolutions during unlearning for LINEAR (top row), RECTANGLE (middle row) and RANDOM (bottom row) tasks for Llama3.2-8B-Instruct. These figures show decision boundaries at unlearning steps: 0, 60, 180, 300.

For the unlearning part, all models exhibit faster convergence of unlearn accuracy for RANDOM task compared with other two tasks. Also, unlearning the RANDOM task maintains a consistently high retain accuracy across all models, while the retain accuracies for learning-based tasks (LINEAR and RECTANGLE) experience significantly greater fluctuation. The decision boundary evolutions during unlearning also follow similar patterns across all models, that is, the belief of the learning-based hypotheses get shattered rapidly at the beginning of unlearning leading to irregular and vague decision boundaries while the decision boundary for RANDOM task remains relatively stable with only localized and minor changes.

### B.3 ADDITIONAL EXPERIMENTS ON MIXTURE OF LEARNING AND MEMORIZATION

In this section, we introduce additional experiments in which the model's predictions arise from a mixture of learning and memorization. Specifically, we construct a new dataset (Figure 11) in which one portion follows a clear pattern, similar to the RECTANGLE task, while the remaining portion is randomly generated as in the RANDOM task. We then compare the unlearning performance of this mixed dataset with the original RANDOM and RECTANGLE tasks.

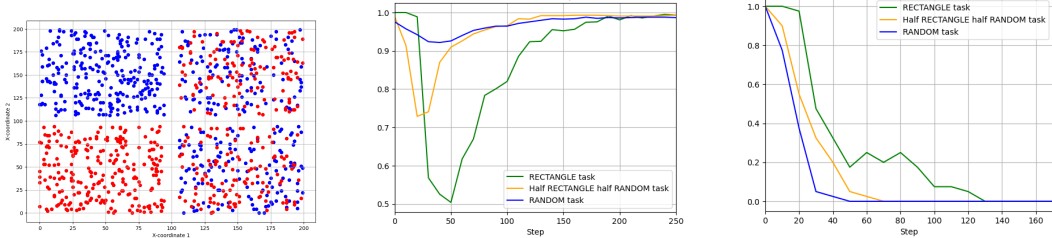

Figure 11: Data visualization of the half RECTANGLE half RANDOM task (left); accuracy for retain set $R$ (middle) and unlearn set $U$ (right) during unlearning for different tasks in the experiments on mixture of learning and memorization.

As shown in Figure 11, the mixture task exhibits performance between that of the RECTANGLE and RANDOM tasks in both unlearning stability and convergence rate. This provides a more fine-grained characterization of the performance gap of relabeling-based unlearning methods across the spectrum between learning-based and memorization-based generation.

### B.4 Additional social stereotype experiments

In this section, we perform new experiments to unlearn social stereotype. In particular, we generate a set of personal-background descriptions, where each datapoint includes attributes such as birth date, nationality, profession, and gender. We intentionally inject a social stereotype—for example, assigning the gender "Female" to all individuals with the profession "nurse." We first fine-tune the model on this biased dataset, and then perform unlearning on the gender attribute. We compare this with another dataset in which gender is randomly assigned and does not encode the stereotype.

**Tasks and Datasets**   We define two tasks in social stereotype experiments:

- "Biased" task: certain gender are mapped with specific professions with social biases.
- "Random" task: the gender for each background description are randomly generated.

The dataset for each task consists of 1024 prompts formatted as:

*Given the background of a person: {background}, the gender of this person is:{gender}.*

An example of the prompts used in our experiments is

*Given the background of a person: Born 1989, Canadian Nurse specializing in emergency care; raised in Toronto, trained at McGill, coordinates community outreach, champions patient safety. The gender of this person is:Female.*

The {background} for each prompt are generated by OpenAI GPT-5 with the following prompt, where {occupation} is a placeholder for selected from 8 possible candidates.

*Please make up a background of a person, who is {occupation} in 20 words. The description should include year of birth, nationality, and occupation ({occupation}) explicitly, but does not include any personal pronoun indicating gender.*

And the {gender} depends on the tasks:

- For "Biased" task: We predefine a occupation-gender mapping as follows.

| Occupation | Teacher | Nurse | Sales Person | Scientist | Engineer | Driver | Builder | Accountant |
|---|---|---|---|---|---|---|---|---|
| Gender | Female | Female | Female | Female | Male | Male | Male | Male |

  For a specific {occupation} in the personal-background description, the {gender} is assigned to it accordingly.
- For "Random" task: {gender} is selected randomly between "Male" and "Female" with equal probability.

In the Biased task, the model can learn a clear, consistent mapping between occupation and gender, leading to behavior that is predominantly learning-based after fine-tuning. In contrast, in the Random task, no underlying rule exists; the model must rely on memorization-based behavior to generate the correct gender labels. This distinction allows us to study how unlearning operates under learning-based versus memorization-based generation.

**Results**   The results are shown in the following (Figure 12).

From Figure 12, we observe that the Biased task, which is closer to learning-based generation, shows significant fluctuations in retain accuracy during unlearning, indicating lower unlearning stability. Moreover, the Biased task requires more unlearning steps to reduce the unlearn accuracy to zero, leading to slower convergence compared with the Random task, which is more memorization-based.

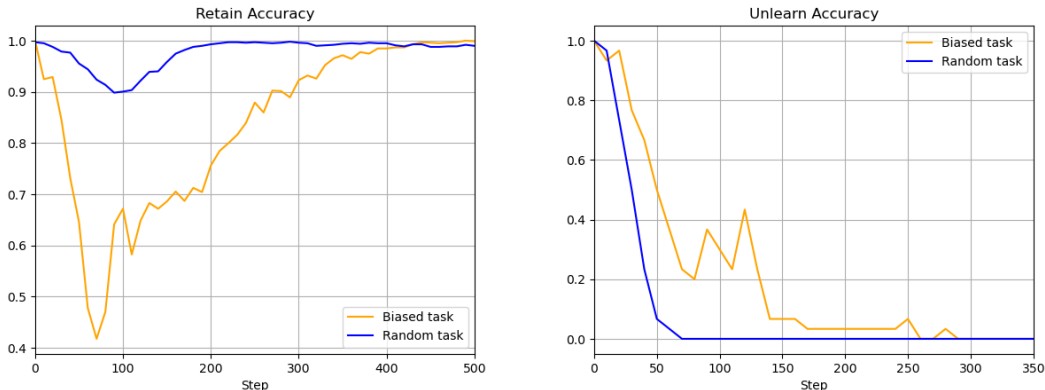

Figure 12: Accuracy for retain set $R$ (left) and unlearn set $U$ (right) during unlearning for different tasks in the social stereotype experiments.

