# OpenReview forum: "The Role of Learning and Memorization in Relabeling-based Unlearning for LLMs"
_ICLR.cc/2026/Conference — Submitted to ICLR 2026_

### Official Review · Reviewer_oVtL · 2025-10-31

**Soundness:** 3
**Presentation:** 3
**Contribution:** 3
**Rating:** 4
**Confidence:** 3

**Summary:**

Authors claim that undesirable outputs can arise either from rule-learning (discovered through training) or memorization (just memorizing individual training instances). The claim is that relabeling-based unlearning works more effectively for overwriting undesirable outputs acquired through memorization, with a Bayesian-theory-based analysis. In high-level terms the proofs provided show that in light of contradictory evidence, decision boundaries shift globally when learning is rule-based, leading to degraded performance on many (potentially irrelevant) inputs. Meanwhile, in the memorization regime, the contradictory evidence only shifts things locally, so relabeling-based unlearning can affect only that fact more easily without touching others. At a high level, relabeling works well for memorization but not for learned rules.

Authors then provide some light experimental evidence of binary 2D Foo/Bar classification, under 3 different label regimes. They show that the 2/3 cases where the model learns a general rule, relabeling-based unlearning is more difficult. In the 1/3 of the cases where the model has to rely on memorization due to the randomness of the labels, unlearning is easier.

**Strengths:**

- the distinction between learning-based and memorization-based generation and how it relates to unlearning is interesting and novel
- the intuition of why learning-based unlearning is more "challenging" for the model than memorization-based unlearning makes sense

**Weaknesses:**

- the experimental setup feels extremely toy-ish. i understand the need to create a constrained scenario but "known function vs. random vector" feels very far from "learning stereotypes vs. memorizing personal information"
- there are unlearning papers that do not deal with such artificial scenarios AFAIK -- why weren't more real-world scenarios used? factual forgetting?

- paper is structured a bit bizarrely -- mostly proofs with the experimental setup seeming to be somewhat of an afterthought, despite feeling like a core part of the argument

- the math/proofs feel very simplified and make a lot of assumptions that make them very rough approximations for modern LLMs in an open-ended generative setting. binary labels, constant-prediction baselines, flip noise, etc. it's very abstracted

**Questions:**

- do you think the learning vs. memorization dichotomy exists for other unlearning methods or just for relabeling?

- learning-based and memorization-based generation in real world terms probably exist as a spectrum, no? not that something cleanly falls in one category or another. would be interesting to provide some sort of method or heuristic to discover where on the spectrum a given dataset lies

---

> ### Author Response · Authors · 2025-11-21
> **Official Comment by Authors**
>
> We thank the reviewer for their thoughtful comments.
>
> **Regarding the comments on the experiments being toy-ish**, we would like to emphasize that our main contribution is theoretical, and the experiments are intended as proof-of-concept validations. Nonetheless, we agree that more realistic settings will further strengthen the paper, and we have added new experiments (Section B.4 in the Appendix) on unlearning social stereotypes, which better reflect real LLM use cases.
>
> In the new experiments, we generate personal-background descriptions with attributes such as birthdate, nationality, profession, and gender, and intentionally inject a stereotype, for example, assigning “Female” to all individuals in the “nurse” profession. After fine-tuning the model on this biased dataset, we perform unlearning on the gender attribute and compare the results to a dataset with randomly assigned gender. Unlearning the stereotype-injected dataset proves significantly more difficult, showing unstable retain accuracy and slower convergence. This aligns with our theoretical prediction that relabeling-based unlearning is more challenging when the underlying behavior is learning-based rather than memorization-based.
>
> **Regarding the comment on applying real-world unlearning scenarios (e.g., factual forgetting)**, our theoretical framework is designed for general prediction settings, with LLMs serving as a key motivating example. However, existing factual unlearning benchmarks are not constructed to distinguish between learning-based and memorization-based knowledge; these components are typically entangled with each other. This makes it difficult to isolate the specific behaviors our theory aims to analyze, and thus challenging to conduct controlled experiments on such benchmarks. Nevertheless, we agree that incorporating broader factual unlearning tasks would further strengthen the work, and we view this as an important direction for future research.
>
> **Regarding the comments about the paper structure**, we agree with the reviewer that the experiments are an important and integral part of this work. However, we want to emphasize that the primary contribution of this work is theoretical where we provide a formal analysis of belief change during unlearning and to establish lower bounds on the challenge of prior change. The purpose of our experiments is to act as a proof of concept that empirically validate the theoretical claims presented in Section 3 that relabeling-based unlearning works better for memorization-based generation than for learning-based generation. Furthermore, our experimental setup closely follows the binary classification setting used in the theoretical analysis, ensuring consistency between the theory and empirical results.
>
> **Regarding the comment on the simplified math proofs and assumptions**, we believe that starting from an abstract and tractable setup, such as binary labels and constant-prediction baselines, is essential for obtaining clear analytical insights into the problem. This simplification allows us to isolate and understand the fundamental mechanisms underlying the relabeled-based unlearning on learning and memorization-based generations.
>
> We also note that, despite its abstract nature, our theoretical model captures aspects of several real LLM applications. For example, QA datasets such as BoolQ [1] and NQ [2] contain substantial portions of binary questions (e.g., “Yes/No,” “True/False”), which naturally fit into our model. That being said, we agree with the reviewer that extending the analysis to more general settings is an interesting direction for future work.
>
> [1] Clark, Christopher, et al. "Boolq: Exploring the surprising difficulty of natural yes/no questions." arXiv preprint arXiv:1905.10044 (2019).
>
> [2] Kwiatkowski, Tom, et al. "Natural questions: a benchmark for question answering research." Transactions of the Association for Computational Linguistics 7 (2019): 453-466.

---

> > ### Author Response · Authors · 2025-11-21
> > **Official Comment by Authors**
> >
> > **Answers to Questions:**
> >
> > 1. **Other unlearning methods**: our analysis relies on the setting of the relabeling-based unlearning method by modeling it as providing the model with fabricated evidence that conflicts its current predictions. Other unlearning methods, such as gradient ascent and negative preference optimization (NPO), do not naturally fit into this modeling framework. As a result, it remains an open question whether these alternative methods would show similar differences in performance between learning-based and memorization-based generations.
> > 2. **Spectrum of learning and memorization-based generations**: we agree with the reviewer that learning-based and memorization-based generation should be viewed as a spectrum rather than a strict dichotomy, and that a model’s behavior often reflects a mixture of both. There are several possible ways to estimate where a dataset or task lies along this spectrum. For example, in our experiments, we observed a correlation between the decision boundary and the model's learning/memorization behaviors: highly structured or global boundaries tend to indicate learning-based behavior, while highly fragmented or local boundaries suggest memorization. Another useful indicator is the model’s uncertainty on unseen data points. Memorization-based generation assumes independence between inputs, leading to high uncertainty on unobserved examples. In contrast, learning-based generation generalizes rules across inputs, resulting in high confidence and low uncertainty for predictions on unseen data. Note that our theoretical results still hold for the case when a model’s behavior depends on a mixture of learning and memorization-based hypotheses. Please refer to our response to Reviewer X47C for more details.

---

### Official Review · Reviewer_d8cB · 2025-10-31

**Soundness:** 4
**Presentation:** 3
**Contribution:** 1
**Rating:** 6
**Confidence:** 4

**Summary:**

This paper investigates what kinds of knowledge can be effectively unlearned by a large language model (LLM). The authors distinguish between two mechanisms of knowledge formation: learning-based, where the model abstracts general rules to approximate data, and memorization-based, where it stores and reproduces specific information verbatim. The study finds that memorization-based knowledge can be unlearned more efficiently through relabeling-based unlearning compared to knowledge derived from generalization.

**Strengths:**

- The paper broadens our understanding by demonstrating that generalizable patterns are significantly harder to unlearn than memorized data points.
- It presents a clear and well-defined thesis that is both theoretically motivated and empirically validated.
- The explanation of the loss function is particularly clear and accessible.
- Well-chosen examples provide strong intuition about the mechanisms underlying unlearning.
- The experimental design is logical and consistent with the theoretical claims, reinforcing the paper’s overall coherence.

**Weaknesses:**

- While the paper highlights important aspects of unlearning, the broader impact of these findings remains unclear. What are the practical or theoretical implications of this additional knowledge? How might it inform future research or applications?
- The finding that general rules are more difficult to unlearn than isolated memorized points is somewhat expected and lacks an element of surprise.
- The motivation for unlearning generalized rules could be made clearer. Under what real-world circumstances would such unlearning be necessary or beneficial? The authors could strengthen their argument by presenting a concrete example where unlearning a general rule is desirable or essential.
- The paper would benefit from a visual summary (e.g., a conceptual diagram) near the beginning to make the core findings more accessible to readers.

**Questions:**

- What do the authors mean by “unstable”? While the general concept is clear, how is instability defined and measured in the context of their framework?
- How does this work relate to the field of alignment? There appears to be a conceptual overlap between unlearning generalized rules and ensuring model alignment. Could the authors clarify whether their findings have implications for alignment research?

---

> ### Author Response · Authors · 2025-11-21
> **Official Comment by Authors**
>
> We thank the reviewer for their positive feedback.
>
> **Regarding the broader impact of our findings**, analyzing the factors influencing the efficiency of relabeling-based unlearning is valuable given the popularity of this approach. Our results show that relabeling-based methods are more effective when the undesirable behavior is based on memorization. In practice, this suggests that the relabeling-based method is particularly suitable for removing memorized information, such as personal information.
>
> At the same time, our analysis provides theoretical insights on why the relabeling is less effective for learning-based generation: it leads to significant belief drops and forces changes to the model’s prior over hypotheses space. This insight motivates the development of new unlearning methods designed specifically to handle learning-based behaviors more efficiently.
>
> **Regarding the comment that our findings appeared expected**, to the best of our knowledge, this is the first work to analyze unlearning difficulty and efficiency from the perspective of learning-based versus memorization-based generation. Although the high-level intuition may appear natural, our contribution is to provide formal analysis via the lens of hypothesis testing and provide theoretical insights underlying the performance gap. We further empirically demonstrate our findings with experiments aligned with our theoretical claims.
>
> **Regarding the comment on the motivation example for unlearning a general rule**, a concrete example of unlearning general rules arises in the context of removing social stereotypes. The model may develop certain biased associations—such as mapping the profession “nurse” to “female” or “engineer” to “male”. These associations reflect generalized rules learned by the model rather than isolated memorized facts. In such cases, it is often necessary to unlearn these stereotypes or biases, making the unlearning of general rules both relevant and important.
>
> To strengthen this motivation, we added a new experiment focused on unlearning intentionally injected social stereotypes, demonstrating the difficulty of removing rule-based generalizations in practice. Please refer to our response to Reviewer X47C for the details and findings of these additional experiments.
>
> **Regarding the comment on the visual summary**, we are happy to follow the reviewer’s suggestion and include a visual summary. To make sure the figure aligns with the reviewer’s expectations, we would appreciate clarification on whether the summary should focus more on the theoretical insights, the experimental observations, or the high-level conceptual relationship between learning- and memorization-based generation under relabeling-based unlearning.
>
> **Answers to Questions:**
> 1. **Definition of "stability"**, the term “stability” reflects how the belief of the hypothesis changes in the presence of contrary evidence, which is defined as \Delta_e in Theorem 2 and 3. We say the memorization-based hypothesis is relatively stable since its belief change is upper bounded even when the evidence length goes to infinity while the learning-based hypothesis scales with the evidence length.
> 2. **Connection to LLM alignment**, this work is closely related to the field of LLM alignment, where part of the objective is to guide the model toward producing desirable responses to certain prompts. Our results suggest that behaviors governed by generalized rules are inherently more difficult to unlearn with relabeling-based methods, which has direct implications for alignment: aligning responses that arise from learning or generalization may be significantly harder than modifying memorized behaviors. We view extending our theoretical framework to alignment settings as an important and promising direction for future work.

---

### Official Review · Reviewer_daWV · 2025-11-01

**Soundness:** 1
**Presentation:** 2
**Contribution:** 2
**Rating:** 2
**Confidence:** 3

**Summary:**

This paper studies how different forms of response generation, learning-based generalization versus memorization of examples, affect the difficulty of machine unlearning. It introduces a Bayesian framework comparing how the posterior belief in these two hypotheses changes when presented with fabricated, conflicting evidence. The theoretical analysis suggests that memorization-based hypotheses are more stable under such adversarial evidence, whereas learning-based hypotheses experience belief collapse as contradictory examples accumulate. The authors then conduct experiments using a single finetuned language model on synthetic  classification tasks and visualize decision boundaries before and after unlearning.

**Strengths:**

1. The paper establishes a controlled way of running experiments to quantify which types of samples are harder to unlearn. This helps articulate an emerging question on how large language models maintain or lose generalization under unlearning interventions. The theoretical setup is presented in an accessible way and highlights a meaningful direction for studying robustness of model beliefs.

2. The theorems show that learning-based hypotheses suffer compounding belief degradation under concentrated noise while memorization-based hypotheses remain stable. Even if the setting is simplified, the formal results help clarify why unlearning strategies might be brittle.

**Weaknesses:**

1. Limited applicability. The datasets are artificially constructed and low-dimensional. Because the tasks are artificially simple and small-scale, and because the unlearning procedure is evaluated only on these toy domains, the conclusions drawn about “learning-based” vs. “memorization-based” behavior do not reliably translate to practical LLM applications. Moreover, it is unclear what is the take-away for better unlearning methods, or if there is metric to check for difficulty when the boundary between learning and memorization is unclear.

2. Gaps in the belief modeling. The hypotheses $H_0$ and $H_1$ are defined only as generative structures for labels, not as full probabilistic models over functions or latent variables with priors. However, in the posterior stability analysis, each hypothesis is assumed to have a well-specified likelihood and a fully Bayesian update mechanism. Since these are not part of the hypothesis definitions, the comparison of posterior collapse reflects assumptions introduced later, not properties implied by the hypotheses themselves. All fabricated evidence uses one input which maximally harms learning models and minimally affects memorization, affecting the conclusions as it is biased towards memorization.

3. Change in assumptions. The paper changes the modeling assumptions mid–analysis: the hypotheses in Theorems 2–3 operate in an unstructured discrete input world with independent labels, but Theorem 4 suddenly invokes structured continuous inputs and linear hypothesis spaces. Because the two regimes have fundamentally different generalization properties, the results about posterior stability and memorization do not carry over. As a result, the conceptual connection claimed between the Bayesian analysis and neural unlearning behavior is unsupported.

**Questions:**

1. Unclear latent: In H1, the hypothesis on memorization, what is the latent vector? How can such hypotheses be tested in practice, does it depend on the input $x$? Do we treat flip noise as part of the latent model?

2. How is the decision boundary estimated for the LLM experiments? The theoretical claims about decision boundary structure do not follow from the underlying model hypotheses which are about outputs ($H_0$) and latent vectors ($H_1$).

3. How is memorization of examples any different from defining exposure as a metric [1], or other works that explore out-of-distribution (random) examples [2], or works studying unlearning hardness [3]?

[1] Carlini, Nicholas, et al. "The secret sharer: Evaluating and testing unintended memorization in neural networks." 28th USENIX security symposium (USENIX security 19). 2019.

[2] Baluta, Teodora, et al. "Unlearning in-vs. out-of-distribution data in LLMs under gradient-based method." arXiv preprint arXiv:2411.04388 (2024).

[3] Zhao, Kairan, et al. "What makes unlearning hard and what to do about it." Advances in Neural Information Processing Systems 37 (2024): 12293-12333.

---

> ### Author Response · Authors · 2025-11-21
> **Official Comment by Authors**
>
> We thank the reviewer for their detailed comments and we are happy to provide further clarifications on our contribution and methods.
>
> **Regarding the comment on limited applicability**, we would like to clarify that the primary contribution of this paper is theoretical. Our focus is on formally analyzing belief change during unlearning and establishing lower bounds on the difficulty of prior adjustment. The experiments are intended as proof-of-concept validations of the theory ​​rather than a comprehensive evaluation on large-scale LLM tasks. That said, we agree that more realistic language-based experiments can further enhance the work. In response, we have added new experiments on unlearning social stereotypes (Section B.4 in the Appendix), which more closely reflect real-world LLM scenarios. Please refer to our response to Reviewer X47C for the details and findings of these additional experiments.
>
> When the decision boundary is unclear, one way to distinguish between learning and memorization generation is on the model’s uncertainty on unseen data. Based on our definitions, memorization-based generation assumes independence between inputs, leading to high uncertainty on unobserved examples. In contrast, learning-based generation generalizes rules across inputs, resulting in high confidence and low uncertainty for predictions on unseen data.
>
> **Regarding the comment on gaps in the belief modeling**, we would like to clarify that both $H_0$ and $H_1$ is a full probabilistic model characterizing the conditional distribution of $y$ given input $x$. In particular, for $H_0$, $P(y=f(x)|x)=1$ for any $x\in [N]$, and for $H_1$, $P(y=1|x)=1/2$ for any $x \in [N]$. Together with the distributional assumptions on input $x$ and observation $z$, this provides a complete distributional characterization for the prompt-response pairs, which justifies the following probabilistic update computation.
>
> **Regarding the comment on the use of the same input for all fabricated evidence**, the choice of using the same input for all fabricated evidence is primarily for analytical tractability rather than introducing bias. Allowing different inputs would involve additional constructions on the baseline hypotheses and significantly complicate the analysis while offering little additional theoretical insight. By focusing on a single-input construction, we are able to cleanly isolate the core mechanisms driving the differing behaviors of learning-based and memorization-based generation under relabeling-based unlearning. We believe this simplification provides a clearer and more interpretable foundation for the theoretical results.
>
>
> **Regarding the comment on changes in assumptions**, we would like to clarify that the change in assumptions across the theorems primarily reflect the distinct questions each theorem addresses. For the input structure, both Theorems 2–3 and Theorem 4 consider discrete input spaces with binary outputs; the primary difference lies in the input dimensionality. However, the results in Theorems 2–3 can be directly extended to higher-dimensional inputs without modification, as the bounds derived there are independent of both the input cardinality N and the input dimension. We will add a remark to the paper to clarify this point.
>
> Furthermore, Theorem 2-3 focus on belief changes for a given function $f$, while Theorem 4 examines the necessity of prior changes over the hypothesis space in the context of learning-based generation.Therefore, the notion of a hypothesis space (linear classifier) does not arise in Theorem 2 and is introduced only in Theorem 4, where it becomes relevant. Moreover, either the belief update shown in Theorem 2-3 or the lower bound results in Theorem 4 does not rely on the generalization property of the learned function.

---

> ### Author Response · Authors · 2025-11-21
> **Official Comment by Authors**
>
> **Answers to Questions:**
> 1. **Regarding the questions of unclear latent**, the latent vector $V$ can be seen as the lookup table or dataset to store the information to be memorized. For example, the medical record number for a patient. Here, patient name is the input $x$ and $y$ is the medical record number of patient $x$. To test this hypothesis, we can compute or estimate the posterior of $P(H_1|h)$ where $h$ is all the observations we have encountered. Note that since $H_1$ completely characterize the conditional distribution of $y$ given $x$, the posterior $P(H_1|h)$ is well defined. The flip noise is defined as the noise between observation z and true response y thus is independent of the latent vector and its sampling.
> 2. **Regarding the question on the decision boundary**, to estimate the decision boundary, we select all points in [0, 200]X[0, 200] grid with a step size of 4 (2500 points in total) and plug it into the prompts shown in Section 4.1 and record the model generation outputs for each point. The resulting patterns are consistent with our hypothesis definitions: learning-based generation leads to smooth, structured decision boundaries, whereas memorization-based generation yields scattered, patternless boundaries. We would greatly appreciate further clarification from the reviewer on the claim that “the theoretical claims about decision boundary structure do not follow from the underlying model hypotheses,” as the observed patterns appear consistent with our theoretical framework.
> 3. **Regarding the comparison with prior works**, our notion of memorization refers to a specific type of model behavior that arises under tasks where the input–output mapping is inherently arbitrary or random. In such settings, no underlying rule or generalizable pattern exists, and the optimal strategy for the model is to memorize the response associated with each input. In contrast, the memorization studied in [1] and [2] is defined through an entropy-based exposure measure—specifically, the reduction in uncertainty about a secret (canary) string after interacting with the model. Similarly, the notion of memorization in [3] corresponds to changes in the model’s output distribution caused by the inclusion of a particular training sample. Our definition is therefore different: it characterizes a model’s reliance on instance-specific lookup behavior induced by the task structure, rather than exposure-driven leakage [1,2] or distributional sensitivity [3] to individual examples.

---

### Official Review · Reviewer_X47C · 2025-11-01

**Soundness:** 2
**Presentation:** 2
**Contribution:** 2
**Rating:** 4
**Confidence:** 3

**Summary:**

This paper examines the effectiveness of relabeling-based unlearning for LLMs, distinguishing between learning-based and memorization-based generation. The results show that relabeling-based unlearning is more effective for memorization-based generation, while learning-based generation proves more resistant to unlearning both in theory and practice. The paper establishes bounds on belief updates, provides a lower bound illustrating the difficulty of changing priors, and empirically confirms these findings on controlled binary tasks. Experiments demonstrate that memorization-like (random) tasks are unlearned faster and with stable retain accuracy, whereas rule-like tasks experience global disruption and slower convergence.

**Strengths:**

- The paper uniquely focuses on the distinction between learning-based and memorization-based generation in LLMs, which is an important aspect of understanding how the model behaves.
- The paper provides formal theoretical analysis, offering bounds on belief updates and illustrating the challenges of changing priors.

**Weaknesses:**

- While the research offers valuable theoretical insights, the empirical section has a limited connection to a practical LLM unlearning scenario. In the experiments, the tasks (classifying 2-dimensional points into binary labels) are quite simple and may not accurately represent how LLMs learn and memorize information in real situations. First, in practice, knowledge in LLMs is learned through sentences both explicitly and implicitly, rather than explicit question prompts like "What is the label for this input?\n Input: 62 87 \n Label:". Second, practical LLM training involves a mix of learning and memorization, not just one or the other as in the experiments. This greatly limits how well the findings apply to more complex, real-world cases. Controlled experiments are certainly acceptable (and even helpful), but the paper could improve by including experiments on more realistic language tasks (such as learning and unlearning factual knowledge) to better demonstrate the practical implications of the theoretical results, since the paper focuses on **LLM unlearning**, not traditional unlearning scenarios for non-LLMs.
- In the empirical section, the paper assumes that if the LLM is given a dataset with regularity (i.e., LINEAR, RECTANGLE), it will learn the underlying rule, whereas if given random data (i.e., RANDOM), it will memorize. However, in practice, LLMs can exhibit a mixture of learning and memorization even on datasets with regularities, depending on factors like model architecture, training regimen, and data complexity. This assumption may oversimplify how LLMs process information, raising questions about the validity of the conclusions drawn from these experiments.
- Minor: The term "hypothesis testing" is used in an uncommon way, which might confuse readers who are familiar with traditional concepts of hypothesis testing. The paper explores the model's unlearning behavior under two hypothetical data generation processes, rather than determining which of the two processes the model functions best under. Clarifying this terminology would improve readability.
- Minor: The paper lacks conclusion and limitation sections. Including these sections would help summarize key findings and discuss the potential limitations of the study.

**Questions:**

N/A

---

> ### Author Response · Authors · 2025-11-21
> **Official Comment by Authors**
>
> We thank the reviewer for their thoughtful and constructive comments.
>
> **Regarding the comment on the limited connection between our empirical section and real LLM unlearning scenarios**, we would like to clarify that the primary contribution of our work is theoretical. Our main focus is to provide a formal analysis of belief change during unlearning using relabeling-based methods and to establish lower bounds on the challenge of prior change. These results are framed for general prediction problems, and LLMs serve as an important motivating application of this theoretical framework. The purpose of our experiments is to act as a proof of concept that empirically validates the theoretical claims, rather than to provide an exhaustive evaluation on full-scale LLM tasks.
>
> That said, we agree with the reviewer that incorporating more realistic language-based experiments would further strengthen the work. In response, we have added new experiments (Section B.4 in the Appendix) on unlearning social stereotypes, which more closely resemble real-world LLM behaviors. Specifically, we generate a set of personal-background descriptions, where each datapoint includes attributes such as birth date, nationality, profession, and gender. We intentionally inject a social stereotype—for example, assigning the gender “Female” to all individuals with the profession “nurse.” We first fine-tune the model on this biased dataset, and then perform unlearning on the gender attribute. We compare this with another dataset in which gender is randomly assigned and does not encode the stereotype.
>
> The results show that unlearning the dataset with intentionally injected social stereotypes is significantly more difficult with unstable retain accuracy and slower convergence compared to the randomly generated dataset. These observations align with our theoretical findings that relabeling-based unlearning is inherently more challenging when the generation is based on learning rather than memorization.
>
> **Regarding the comment on the mixture of learning and memorization in LLMs**, we agree with the reviewer that many real-world tasks require both types of behavior. We would like to emphasize, however, that our theoretical results do **not** assume the model relies solely on learning-based or solely on memorization-based generation. Our framework accommodates the setting in which the model’s prediction depends on a mixture of both.
>
> In particular, the model may maintain multiple hypotheses $H_1,H_2,…,H_k$ with associated beliefs $P(H_i|h)$, where $h$ denotes the history of data observed by the model so far. Some hypotheses may be learning-based, while others may be memorization-based. The model’s prediction is then a mixture over these hypotheses following the law of total probability:
> $$P(y|x, h) = \sum_{i=1}^k P(H_i |h)P(y|H_i, x, h)$$
> Our theoretical results still hold under the setting in which predictions arise from a combination of learning and memorization. Depending on the belief associated with each hypothesis, the model generation may lean towards learning or memorization. If helpful, we can add this as a corollary or remark following the hypothesis definitions.
>
> That said, the goal of this paper is to analyze the performance gap of relabeling-based unlearning methods between the learning and memorization-based generation. Therefore, in our experiments, we intentionally construct datasets in which the belief in particular hypotheses is made prominent, allowing us to cleanly observe and compare the behaviors predicted by our theory. In addition, we include new experiments (Section B.3 in the Appendix) in which the model’s predictions arise from a mixture of learning and memorization by constructing a dataset as a combination of the RECTANGLE (learning-based) and RANDOM (memorization-based) tasks. The performance of this mixture dataset lies between the RECTANGLE and RANDOM tasks in both unlearning stability and convergence rate, providing a more fine-grained characterization of how relabeling-based unlearning methods behave across the spectrum from learning-based to memorization-based generation.
>
> **Regarding the comment on the term “hypothesis testing”**, we apologize for the oversight in using the term. We will add more clarification in the paper to avoid confusion.
>
> **Regarding the comment on the conclusion and limitation sections**, we will add a section of conclusion and limitation to the paper copied as follows.

---

> > ### Author Response · Authors · 2025-11-21
> > **Official Comment by Authors**
> >
> > **Conclusion**:
> >
> > In this paper, we investigate how the nature of a model’s generation influences the efficiency of relabeling-based unlearning, with a focus on distinguishing between learning-based and memorization-based generations. Our results show that relabeling-based methods are more effective for unlearning memorization-based generation, exhibiting more stable belief updates and requiring no significant changes to the model’s priors. In contrast, unlearning learning-based generation is inherently more challenging. Promising future directions include extending our theoretical framework to settings with more general types of response other than binary classification, demonstrating our results on a broader range of benchmarks, and exploring connections to LLM alignment, which shares conceptual similarities with relabeling-based unlearning.

---

### Author Response · Authors · 2025-11-28
**General Response**

We sincerely appreciate the time and effort invested by all reviewers in evaluating our work. Your insightful feedback has been invaluable in strengthening the clarity and impact of our work. Below, we summarize the key revisions and clarifications made in response to the collective concerns, followed by detailed point-to-point responses to each reviewer.

**Enhanced Empirical Validation (Addressing Reviewers X47C, daWV, oVtL):**
- **New Experiment (Rule-Based Bias)**: We expanded our empirical study by adding new experiments on unlearning social stereotypes. This task requires the LLM to unlearn a generalized rule (for example, mapping the gender “Female” to individuals with the profession “nurse"), providing a more realistic and practical application of our theory. The results confirm our theoretical findings that unlearning rule-based knowledge is significantly harder and less stable than unlearning memorized data. (See Appendix B.4).
- **New Experiment (Hypothesis Mixture)**: To characterize the spectrum between learning and memorization, we added experiments using a mixture dataset (RECTANGLE and RANDOM data combined). The results provide fine-grained characterization of how relabeling-based unlearning methods behave across the spectrum from learning-based to memorization-based generation. (See Appendix B.3).

**Theoretical and Conceptual Clarification (Addressing Reviewers X47C, daWV,  d8CB, oVtL):**
- **Clarification on Contribution and Motivation**: We emphasize our primary contribution as being theoretical, offering a formal analysis of belief change during unlearning and establishing lower bounds on the challenge of prior change. We added a concrete, real-world example (unlearning social stereotypes) to motivate why unlearning general rules is relevant and important.
- **Mixture Hypothesis Clarification**: We explicitly clarify that our theoretical analysis supports the setting of mixed hypotheses, where a model's prediction depends on a combination of learning- and memorization-based factors, validating the learning and memorization spectrum viewpoint.
- **Modeling and Assumptions**: We clarified that our hypothesis modeling constitutes a full and well-defined probabilistic model that provides a complete distributional characterization for the prompt-response pairs. Furthermore, we clarified that the change in assumptions across the theorems reflects the distinct questions each theorem addresses, rather than a gap in modeling.

**Structural and Editorial Modification (Addressing Reviewers X47C, d8cB):**
- **Conclusion and Limitations:** We added a Conclusion section to summarize our findings, discuss implications, and highlight promising future directions.
- **Terminology**: We provided clearer definitions for terms like "stability" and clarified the use of "hypothesis testing" to avoid confusion.

We again thank the reviewers for their thoughtful and constructive feedback. We welcome any additional questions and are glad to provide further clarification as needed.

---

### Meta-Review · Area_Chair_KPSV · 2025-12-09

**Summary:**

The paper offers a clear distinction between learning-based and memorization-based generation and presents a Bayesian analysis that explains why relabeling unlearning more readily removes memorized facts than generalized rules. However, the empirical setup is far from practical LLM unlearning scenarios, so the conclusions have limited external validity. Key modeling assumptions shift across proofs and the link to real-world applications and stronger baselines is weak, leaving unclear guidance for practical unlearning methods.

**Reviewer Concerns:**

Addressed: The rebuttal adds more realistic experiments on unlearning social stereotypes and a mixed “learning–memorization” setting, reinforcing the spectrum view and the core theoretical claims. It clarifies the paper’s primary theoretical contribution, details a full probabilistic modeling stance, and explains why different theorem assumptions target distinct questions. It also improves structure with a Conclusion and clearer terminology for “stability” and “hypothesis testing.”

Outstanding. External validity remains limited without broader real-world LLM unlearning benchmarks and stronger baselines. The empirical link between the theoretical bounds and end-to-end LLM behavior needs deeper stress tests and sensitivity analyses. Practical guidance for diagnosing where a dataset lies on the learning–memorization spectrum, and for operationalizing instability in applied settings, is still under-specified.

**Reviewer Scores:**

It is hard to confirm since there weren't discussions between the reviewers and the authors. However, X47C or daWV might have raised their scores if their concerns were indeed addressed by the author responses (unconfirmed).

---

### Decision · Program_Chairs · 2026-01-26

Reject